# Diversity and spatiotemporal variations in bacterial and archaeal communities within Kuwaiti territorial waters of the Northwest Arabian Gulf

Saja A. Fakhraldeen[1]*, Sakinah Al-Haddad[1], Nazima Habibi[2], Surendraraj Alagarsamy[1], Sabeena F. K. Habeebullah[1], Abdulmuhsen K. Ali[2], Walid M. Al-Zakri[1]

1 Ecosystem-based Management of Marine Resources Program, Environment and Life Sciences Research Center, Kuwait Institute for Scientific Research, Salmiya, Kuwait, 2 Biotechnology Program, Environment and Life Sciences Research Center, Kuwait Institute for Scientific Research, Shuwaikh, Kuwait

* sfakhraldeen@kisr.edu.kw

**Data Availability Statement:** All fastq files are available from the NCBI database (bacterial sequences SRA: SUB12314807; Bioproject:

## Abstract

Kuwaiti territorial waters of the northwest Arabian Gulf represent a unique aquatic ecosystem prone to various environmental and anthropogenic stressors that pose significant constraints on the resident biota which must withstand extreme temperatures, salinity levels, and reducing conditions, among other factors to survive. Such conditions create the ideal environment for investigations into novel functional genetic adaptations of resident organisms. Firstly, however, it is essential to identify said organisms and understand the dynamic nature of their existence. Thus, this study provides the first comprehensive analysis of bacterial and archaeal community structures in the unique waters of Kuwait located in the Northwest Arabian Gulf and analyzes their variations with respect to depth, season, and location, as well as their susceptibility to changes in abundance with respect to various physicochemical parameters. Importantly, this study is the first of its kind to utilize a shotgun metagenomics approach with sequencing performed at an average depth of 15 million paired end reads per sample, which allows for species-level community profiling and sets the framework for future functional genomic investigations. Results showed an approximately even abundance of both archaeal (42.9%) and bacterial (57.1%) communities, but significantly greater diversity among the bacterial population, which predominantly consisted of members of the Proteobacteria, Cyanobacteria, and Bacteroidetes phyla in decreasing order of abundance. Little to no significant variations as assessed by various metrics including alpha and beta diversity analyses were observed in the abundance of archaeal and bacterial populations with respect to depth down the water column. Furthermore, although variations in differential abundance of key genera were detected at each of the three sampling locations, measurements of species richness and evenness revealed negligible variation (ANOVA $p < 0.05$) and only a moderately defined community structure (ANOSIM $r^2 = 0.243$; $p > 0.001$) between the various locations. Interestingly, abundance of archaeal community members showed a significant increase ($\log_2$ median ratio of RA = 2.6) while the bacterial population showed a significant decrease ($\log_2$ median ratio = -1.29) in the winter season. These

PRJNA910671; accession number(s) SAMN32137957 – SAMN32137992; URL: https://www.ncbi.nlm.nih.gov/sra/PRJNA910671).

**Funding:** SAF received funding under project code FM120K from the Kuwait Institute for Scientific Research (https://www.kisr.edu.kw/en/). The funders had no role in study design, data collection and analysis, decision to publish, or preparation of the manuscript.

**Competing interests:** The authors have declared that no competing interests exist.

findings were supported by alpha and beta diversity analyses as well (ANOSIM $r^2 = 0.253$; $p > 0.01$). Overall, this study provides the first in-depth analysis of both bacterial and archaeal community structures developed using a shotgun metagenomic approach in the waters of the Northwest Arabian Gulf thus providing a framework for future investigations of functional genetic adaptations developed by resident biota attempting to survive in the uniquely extreme conditions to which they are exposed.

## Introduction

Marine ecosystems are among the most diverse and dynamic ecosystems on earth. The role of various marine microbes in mediating and regulating the healthfulness status of the surrounding marine biota and in the maintenance of essential biogeochemical processes underscores the importance of increasing our understanding of the structure of marine microbial communities [1, 2]. Components of marine ecosystems can vary widely with respect to location, depth, season, physicochemical parameters of the seawater and surrounding environment, etc. [3–6]. Given the limitations of existing technologies, the process of systematically isolating and classifying every component within a given volume of seawater collected under various conditions is a technically demanding task [7, 8]. However, an alternative approach under the umbrella of environmental metagenomics, which involves comprehensive community profiling of aggregate samples has proved to be productive [9]. Furthermore, a cost-benefit analysis of such aggregate analyses revealed a lucrative relationship between the relatively small amount of experimental preparatory work required at the front end and the very large amount of data generated as a result [10].

Thus, marine metagenomics has been successfully applied to catalog aquatic microbial communities in multiple water bodies around the world [1, 6, 9, 11–13]. Additionally, scientists have employed this approach to provide spatially and/or temporally refined environmental microbial community profiles [11, 14]. Moreover, given the tractability of the approach, researchers have applied it to study the response of microbial community structures to various environmental and anthropogenic stressors, such as global warming and discharges from desalination and/or power plants, respectively [15, 16].

Metagenomic microbial community profiling can be performed by sequencing specific marker genes, most commonly the 16S or 18S rRNA genes, or by performing more universal shotgun metagenomic sequencing. 16S rRNA-based profiling often does not allow for identification beyond a genus level [17]. Conversely, shotgun metagenomic analysis can provide high-resolution taxonomic profiling up to the species level, in addition to insights into gene function, metabolism, and functionally relevant polymorphisms that are not accessible from analysis of individual marker genes. Shotgun metagenomic sequencing also provides superior statistical power compared to 16S rRNA sequencing approaches [17]. Metagenomic profiling using a shotgun metagenomics approach permits prediction of individual community- and ecosystem-level responses to environmental disturbances via a tractable and simple workflow [18]. Importantly, establishment of a comprehensive profile of marine microbial communities within a given water body will provide the baseline information required to generate *in silico*-based prediction models, which will be invaluable for developing real time response criteria for local adverse events, such as mass death and disease of marine biota, harmful algal blooms, introduction of anthropogenic stressors, and global phenomena such as climate change [15, 16, 19–21].

Kuwait Bay (850 km$^2$), which is part of the Arabian Gulf and is an extension of the Indian Ocean in the North, is surrounded by the landmasses of the Islamic Republic of Iran and the

Arabian Peninsula. Apart from being an important regional nursery ground for several species of fishes and shrimps, it is a water body of vast environmental, economic, and cultural significance to the State of Kuwait and to the region. The waters of Kuwait Bay are significantly affected by global ocean acidification events [22, 23] as well as other environmental and anthropogenic stressors in the form of microplastics [24, 25], dust [14], naturally occurring radioactive material [26, 27], metals [28], and wastewater [29, 30]. Wastewater discharge introduces a significant influx of antibiotics and other pharmaceutical agents into the marine environment of Kuwait [29, 30]. These factors and other such as salinity, ultraviolet radiation, proximity to oil rigs and desalination plants collectively play an important role in shaping the microbial community composition within these waters. Therefore, this study aimed to address the unmet need of systematically profiling the composition of bacterial and archaeal communities in the waters of the northwest Arabian Gulf using a shotgun metagenomics approach. The results from this study provide a comprehensive baseline for marine microbial community composition within Kuwait Bay and the adjoining areas. Marine microbial community profiles and their variations with respect to time, location, water column depth, and physical characteristics of the surrounding water have been documented via shotgun metagenomics.

## Materials and methods

### Site description and sample collection

Seawater samples were collected from three different locations (stations 18, A, and K6) distributed throughout the northwest Arabian Gulf in the area containing and surrounding the water body known as Kuwait Bay (Fig 1). The selected stations were strategically distributed throughout the study area, with one station located within Kuwait Bay (station K6), one to the northeast of Kuwait Bay (station A), and one to the southeast of Kuwait Bay (station 18) (Table 1). Seawater samples were collected monthly for a total period of six months (September 2019–February 2020) from both the surface and bottom of the water column at average depths of 1.29 m and 11.54 m, respectively (S1 and S2 Tables and S1 Fig). Pre-sterilized 5 L polypropylene carboys (Azlon® cat# BNP05B) were used to store the collected samples, which themselves were approximately 3 L in volume. Samples were chilled in an icebox (4˚C) between the time of sampling and the time of delivery to the laboratory for processing. Collectively, a total of 36 seawater samples (S1 Table) were collected and processed.

### Metadata acquisition

Metadata related to various physicochemical parameters of the seawater and the surrounding air at the time of sampling were also collected and logged (Table 1, S2 and S3 Tables, and S1 and S2 Figs). The measured parameters related to seawater included: $O_2$ levels (ml/L; levels of dissolved oxygen in the seawater), salinity (practical salinity units (psu); salt concentration in the seawater), and seawater temperature (˚C), all of which were measured using a portable water quality profiler (conductivity, temperature, and depth (CTD) profiler) equipped with precalibrated sensors (AAQ-RINKO, JFE Advantech, Japan), sampling depth (m; depth at which samples were collected), total depth (m; total depth of station location), wave height (m; wave height was measured using pressure-based wave and tide acoustic sensors), sea color assessed using the Forel and Ule scales (https://forelulescale.com), and secchi disk (m; measure of water transparency based on depth). The measured parameters related to air included: wind direction, wind speed (kph), RH% (relative humidity measured using a humidity sensor), and barometric pressure (hectopascals; also known as atmospheric pressure), all of which were assessed/measured using a portable weather station (Kestrel), air temperature in shade (˚C), and cloud cover (based on visual assessment).

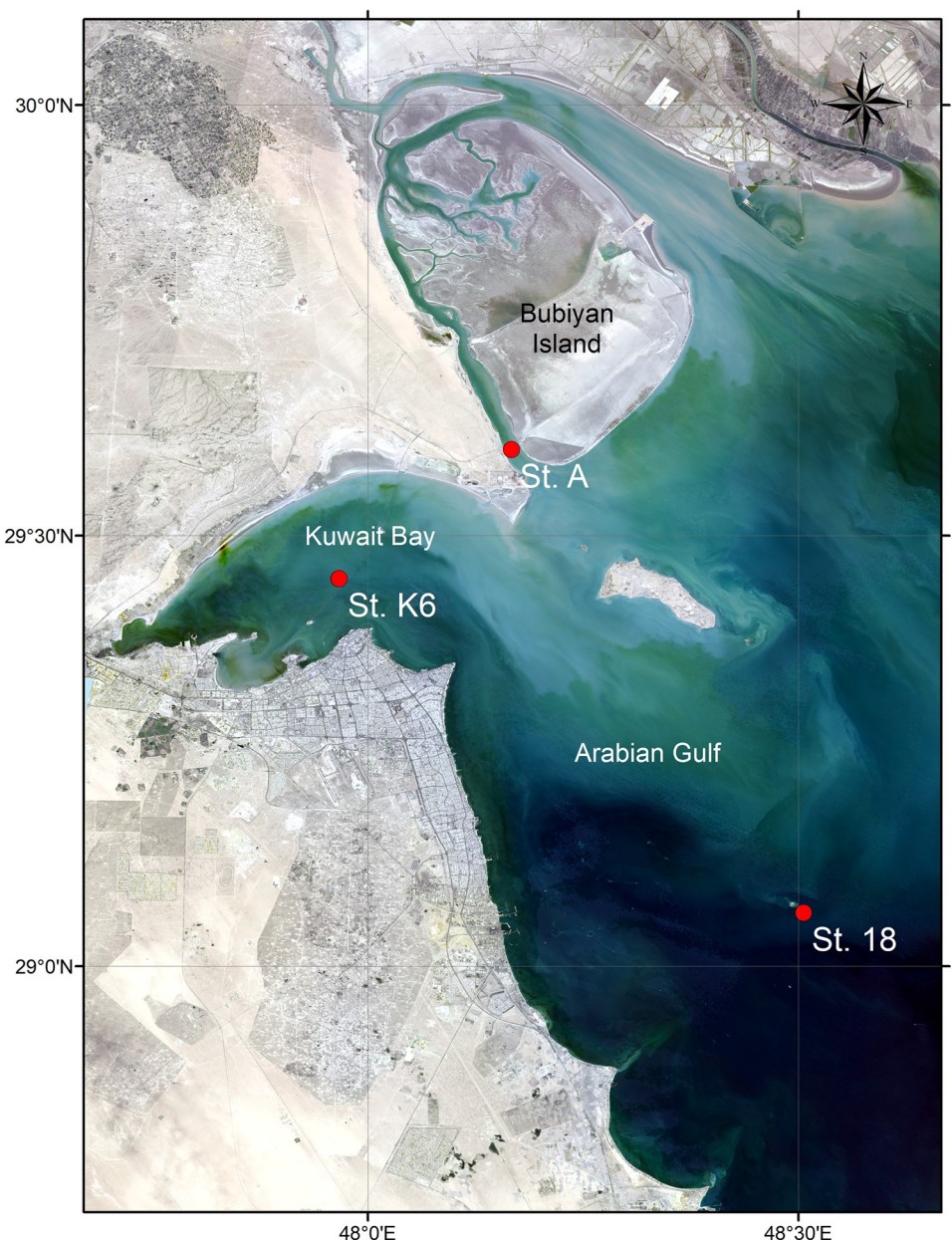

**Fig 1. Sampling site description.** Map showing northwest region of the Arabian Gulf bordering Kuwait City, specifically the water body known as Kuwait Bay (basemap obtained from Landsat (http://landsat.visibleearth.nasa. gov/) and annotated using the ArcGIS software (https://www.arcgis.com/). Red dots indicate the three sampling stations utilized for this study (Station A to the northeast of Kuwait Bay, Station K6 located inside Kuwait Bay, and Station 18 located to the southeast of Kuwait Bay). Figure courtesy of T. Yamamoto, Kuwait Institute for Scientific Research.

## DNA isolation, quality control, and library preparation

Samples were collected in presterilized 5 L polypropylene carboys (Azlon® cat# BNP05B). Immediately upon sample delivery to the laboratory, approximately 1 L of each sample was filtered through a 0.22-μm polycarbonate membrane (Isopore/Millipore cat# GTTP040700) using a vacuum pump apparatus. Filter membranes were then cut using presterilized razor

**Table 1. Location and depth of sampling stations and physicochemical characteristics of the seawater at each station over the six-month sampling period\*.**

| Station | GPS Coordinates | Average measured station depth (m) | Average seawater temperature (˚C) | Average salinity (psu) | Average dissolved oxygen levels (ml/L) |
|---|---|---|---|---|---|
| 18 | 29˚ 03.743 N 48˚ 30.320 E | 20.5 | 25.39 | 40.66 | 6.23 |
| A | 29˚ 36.013 N 48˚ 10.004 E | 6.5 | 21.59 | 40.07 | 7.2 |
| K6 | 29˚ 27.006 N 47˚ 57.990 E | 10.5 | 23.01 | 41.40 | 6.68 |

\*GPS coordinates shown were from the September 2019 sampling excursion. There was minimal variation in coordinates registered for each station between sampling months. A complete list of GPS coordinates per month is provided in S1 Table.

Physicochemical measurements shown are an average of values measured monthly over the six-month sampling period from both the surface and the bottom of the water column at each station. A complete list of monthly values for each of the physicochemical parameters shown and others is provided in S2 and S3 Tables.

The average combined concentration of nitrate + nitrite in water samples from these station was previously reported to be between 0.88–9.82 μM [31].

blades or scissors, and the cut membrane pieces were used as the starting material for DNA extraction. DNA was isolated from the seawater samples using the MO BIO PowerSoil® DNA Isolation Kit (Qiagen) as per manufacturer's instructions. The quality of the extracted DNA was assessed using gel electrophoresis. Specifically, samples were run on a 1% agarose gel containing 10 mg/ml ethidium bromide in an electrophoresis chamber filled with 1X TAE buffer, and 5μl of each sample was mixed with 1μl of loading dye, respectively (Promega cat# G190A) and loaded into the wells. The samples were then run at 100 V for 30 to 90 min. Results were imaged using the ChemiDoc MP gel documentation system (Bio-Rad). The quantity (ng/μl) and quality (260/280 and 260/230 ratios as primary and secondary assessments of nucleic acid purity, respectively) of the DNA yield were also assessed using a NanoDrop™ device (Thermo-Fisher Scientific).

## Shotgun metagenomic sequencing and analysis

A total of 1 ng of DNA was used for library preparation using the Nextera XT DNA Library Preparation Kit (Illumina) as per manufacturer's instructions. Final library quantity was assessed using the Qubit 2.0 system (ThermoFisher, Massachusetts, USA) and quality was checked using the Tape Station 2200 system coupled with high sensitivity D1000 Screen Tapes (Agilent Technologies, Inc., California, USA). Equimolar pooling of libraries was performed based on quality control (QC) values and prepared libraries were subjected to metagenomic sequencing using the HiSeq X platform (Illumina) with read lengths of 2 x 150 bp at a depth of 20M paired-end reads per sample (10M for each of forward and reverse reads). The raw reads were then preprocessed and trimmed for quality, and the adapter sequences were removed using Trimmomatic tool (v0.38) [32] applying the default parameters. The FastQC tool (v0.11.8) (Babraham Bioinformatics 2010, https://www.bioinformatics.babraham.ac.uk/projects/fastqc/) was used to check the quality of raw reads. MEGAHIT (v1.2.9) [33, 34] was used to assemble the trimmed FASTQ data and generate contigs of average length around 640 bp.

## Microbial community composition analysis

The computational tool MetaPhlAn (v3.0) [17] was used to generate taxonomic profiles of microbial community composition at the family, genus, and species levels from the assembled sequences [35, 36]. The mean relative abundance, variance, standard deviation, p-value (based

on a non-parametric t-test, which provides a measure of asymptomatic significance), and q-value (a measure of the minimum false discovery rate at which significance may be assigned to a given test) were calculated. The top 12 differentially abundant taxa at the family-, genus, and species-levels were presented on bar charts. Differential heat trees using the Wilcoxon matched-pairs signed rank/Mann-Whitney test were constructed using the R (v4.0.5) package Metacoder (v0.3.4) [37]. Differential abundance analysis based on a non-parametric t-test was performed using Metastats [38]. Four alpha diversity indices were calculated (Chao1, Shannon, Simpson, and Inverse Simpson (InvSimpson)) using the R package phyloseq (v1.34.0) [39]. P-values for alpha diversity were calculated based on the analysis of variance (ANOVA) statistical test, which was performed using the R function aov in the R package stats4 (v4.0.5). Calculations for the Principal Component Analysis (PCA) were performed using the R function prcomp in R package stats4 (v4.0.5) [40] by singular value decomposition of the centered and scaled data matrix. 3D scatter plots were generated using scatterplot3d (v0.3–41) [41]. Analysis of Similarity (ANOSIM) was performed using the R function "anosim" in the package vegan (v2.5–7) (https://cran.r-project.org/web/packages/vegan/index.html). The R package microbiome_1.13.10 was used for the core microbiome analysis. The function plot_core was used to plot the figures. The minimum prevalence was set to 0.1.

### Correlation between microbial community structure and metadata

Correlation between three of the physicochemical parameters that were measured (seawater temperature, salinity, and dissolved oxygen levels) and the OTU abundance up to the species level was calculated using the Kendall and Spearman methods. The function cor.test from the R package stat (v4.0.5) was used. Kendall's tau and Spearman's rho statistics were used to estimate a rank-based measure of association. For the Spearman test, p values are computed using algorithm AS 89 by default.

## Results

### Metagenomic sequencing and assembly

Paired end sequences (150 x 2) were generated with an average of 15M paired ends reads per sample (30M reads total; Phred > Q20 for 99% of reads). Following quality filtering and trimming, 77% of reads were retained yielding approximately 23M processed reads per sample, or 23M clean reads per sample for analysis. Each sample was *de novo* assembled into scafftigs ranging in length from 136,466,319 to 433,344,004 bp. The N50 of the assembled genomes ranged from 571 to 811 and the lengths of the smallest and largest scafftigs were 37,426 and 515,750, respectively.

### Taxonomic community profiling and core microbiome analysis

Comprehensive taxonomic profiling of bacterial and archaeal populations within all 36 project samples were performed up to the species level. The most abundant taxa were presented in the form of stacked bar plots and heat trees.

The relative abundance (RA) of bacterial and archaeal communities were 57.1%, and 42.9% respectively. The bacterial domain also displayed greater diversity amongst its constituents compared to the archaeal domain.

The most prominent bacterial phylum in the study samples was Proteobacteria (55.4%), which was followed by the phyla Cyanobacteria (32.8%) and Bacteroidetes (11.8%). At the class level, the classes Alphaproteobacteria (74.8%) and Gammaproteobacteria (25.2%) were highly prevalent. Predominant orders in the class Alphaproteobacteria were Rhodobacterales

(66.0%), Rhodospirillales (23.7%), Sphingomonadales (7.1%), and Pelagibacterales (3.1%). Whereas Alteromonadales (93.4%), Cellvibrionales (4.2%), Salinisphaerales (1.3%), and Nevskiales (1.1%) were more prevalent orders in the class Gammaproteobacteria. Among Cyanobacteria, Synechococcales, (99.998%) and Oscillatoriales (0.002%) were dominant. Further classification revealed 12 families to be prevailing in Kuwaiti territorial waters. The eight families with the highest RA were Candidatus_Heimdallarchaeota (42.4%), Synechococcaceae (18.7%), Rhodobacteraceae (14.0%), Alteromonadaceae (7.3%), Flavobacteriaceae (5.7%), and Geminicoccaceae (5.6%), Sphingomonadaceae (0.011%), and Microcoleaceae (0.0004%) (S1 File).

At the genus level, the constituents with the highest RA were unclassified *Candidatus_Heimdallarchaeota* (42.4%) followed by *Synechococcus* (18.7%), *Rhodobacteraceae_unclassified* (13.5%), *Alteromonas* (7.0%), *Geminicoccus* (5.6%), *Formosa* (3.7%), *Trichodesmium* (0.0004%), and *Marinovum* (0.0003%) (S2 File). The most abundant species was the archaeal species *Candidatus_Heimdallarchaeota_archaeon* (42.4%), followed by *Synechococcus_sp_WH_8109* (18.6%), *Rhodobacteraceae_bacterium_HIMB11* (13.0%), *Alteromonas_macleodii* (7.0%), *Geminicoccus_sp* (5.6%), and *Formosa_sp_Hel3_A1_48* (3.7%), *Trichodesmium_erythraeum* (0.0004%), and *Marinovum_sp* (0.0003%) (Fig 2 and S3 File).

The Domain Archaea was represented by two phyla: predominantly Candidatus_Heimdallarchaeota (98.8%) and Euryarchaeota (1.2%). Further downstream interrogation of members of both phyla revealed taxons belonging to unclassified classes, orders, families, and genera (Fig 2b).

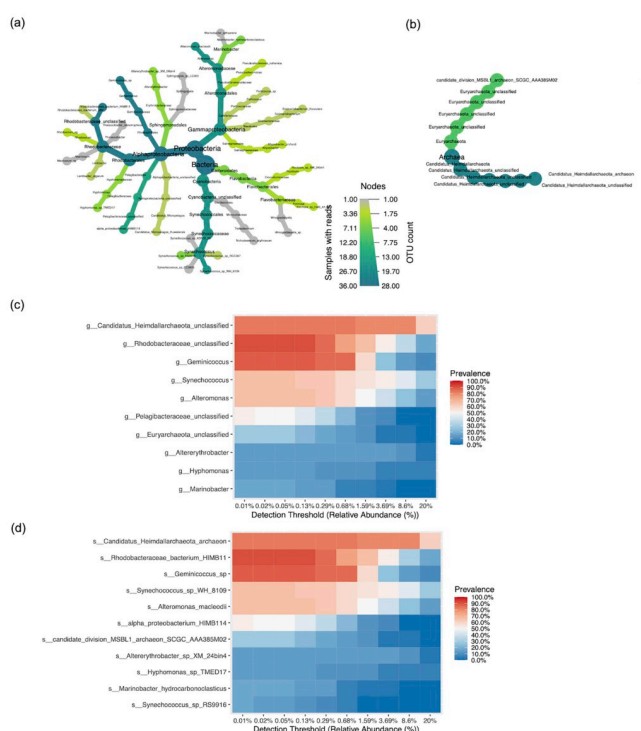

**Fig 2. Taxonomic distribution of bacterial and archaeal populations in the waters of the northwest Arabian Gulf.** Species-level heat trees of (a) bacteria and (b) archaea. The size and color of the nodes are mapped relative to observed read numbers and OTU counts, respectively. Core microbiome analysis at the (c) genus level and (d) species level.

The core microbiome analysis revealed the prevalence (presence in the number of samples) of all the classified genera and species in the marine samples (n = 36) collected from the northwest Arabian Gulf in the current investigation. The top 10 and 11 genera (Fig 2c) and species (Fig 2d), respectively showing prevalence above 0.1 are presented on the heat maps. Maximum percent prevalence at the genus level was shown by *Rhodobacteraceae_unclassified* (14.41%), *Synechococcus* (14.85%), *Geminicoccus* (13.97%), *Candidatus_Heimdallarchaeota unclassified* (13.10%), *Alteromonas* (10.48%), *Pelagibacteraceae unclassified* (8.30%), Euryarchaeota unclassified (4.37%), *Marinobacter* (3.06%), *Altererythrobacter* (2.18%), and *Hyphomonas* (2.18%). Maximum prevalence at the species level was shown by *Rhodobacteraceae bacterium_HIMB11* (14.41%), *Geminicoccus sp* (13.97%), *Candidatus Heimdallarchaeota archaeon* (13.10%), *Synechococcus sp WH 8109* (10.48%), *Alteromonas macleodii* (10.48%), *alpha proteobacterium HIMB114* (8.30%), *candidate division MSBL1 archaeon SCGC AAA385M02* (4.37%), *Synechococcus sp RS9916* (2.62%), *Marinobacter hydrocarbonoclasticus* (2.62%), *Hyphomonas sp TMED17* (2.18%), and *Altererythrobacter sp XM 24bin4* (2.18%). The remaining genera and species had prevalence below 0.1 (F). A close look at the abundance heat trees further demonstrates the predominance of these genera and species (S4–S6 Files).

## Effect of sampling depth on bacterial and archaeal populations

As the samples were collected from both the surface (average depth 1.29 m) and the bottom (average depth 11.54 m) of the water column, variations in overall abundances of bacterial and archaeal communities with respect to depth were speculated. As expected, differences in the RA of predominant taxa at all the taxonomic levels were observed. The lower taxonomic levels, being the active components of biogeochemical processes, were studied in more details. The abundances were compared using the Wilcoxon matched-pairs signed rank/Mann-Whitney test on the RA at p<0.05 (S7 File). The top twelve taxa were presented on the bar charts that revealed family *Alteromonadaceae* (Fig 3a), genus *Alteromonas* (Fig 3b), and its member

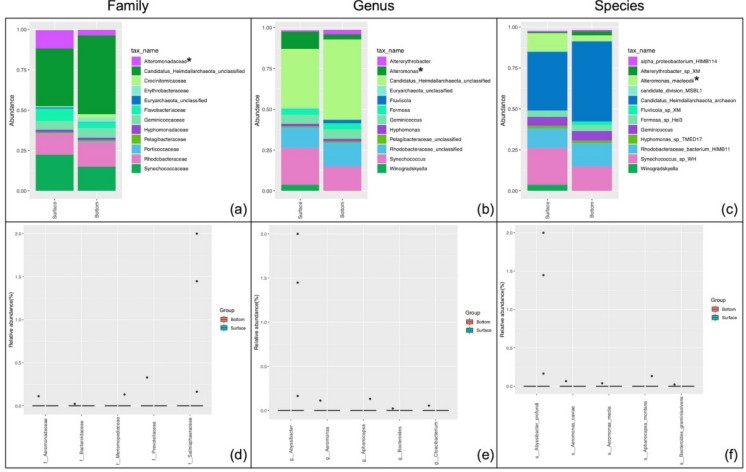

**Fig 3. Comparison of taxonomic distribution of bacterial and archaeal populations relative to sampling depth (surface (n = 18) versus bottom (n = 18)).** Differential abundance analysis employing Wilcoxon matched-pairs signed rank/Mann-Whitney test at the (a) family, (b) genus, and (c) species level. The relative abundance of the twelve most abundant taxonomic groups per classification are depicted. *p<0.02 differential abundance testing through a non-parametric t-test at an adjusted q value <0.05 at the (d) family, (e) genus, and (f) species level. The displayed taxa have a q value <0.009 and p value <0.001.

species *Alteromonas macleodii* (Fig 3c) to vary significantly in surface versus bottom samples (p = 0.01).

The differential abundance analysis (adjusted false discovery rate (q value)) was also performed to compare the most differentially abundant taxa between surface and bottom samples. In total 6, 9 and 18 families, genera, and species were significantly different at q < 0.05 (S8–S10 Files). The top five are presented in Fig 3d and 3e. Families Aeromonadaceae (q = 0.009), Bacteroidaceae (q = 0.009), Merismopediaceae (q = 0.009), Prevotellaceae (q = 0.009), Salinisphaeraceae (q = 0.009), and Sinobacteraceae (q = 0.009) exhibited significantly differential abundance between the surface and bottom of the water column (Fig 3d). Similarly, the genera *Abyssibacter*, *Aeromonas*, *Bacteroides*, *Cloacibacterium* and *Aphanocapsa* were different at q<0.05 (Fig 3e). The predominant significantly different species were *Abyssibacter profundi* (q = 0.0048), *Aeromonas caviae* (q = 0.0048), *Aeromonas media* (q = 0.0048), *Aphanocapsa montana* (q = 0.0048), and *Bacteroides graminisolvens* (q = 0.0048) (Fig 3f).

Regarding the archaeal communities the unclassified archaeal classes belonging to the phyla Candidatus_Heimdallarchaeota and Euryarchaeota displayed non-significant differential abundance between surface and bottom samples.

## Spatial variations in community profiles

Samples were collected from three different stations located in and around the shallow water body of Kuwait Bay in the northwest region of the Arabian Gulf. Each location was geographically and environmentally distinct. Therefore, we sought to investigate the presence of any accompanying spatial variations in the bacterial and archaeal populations residing in these waters.

Overall, both archaeal and bacterial community profiles showed significantly different abundances between stations 18 and A (p<0.02), with the archaeal population showing more abundance at Station A (log2 median ratio of RA = -1.67) while the bacterial population displayed significantly higher abundance at station 18 (log2 median ratio = 1.71). The archaeal phylum Candidatus_Heimdallarchaeota was significantly preferential at Station A (p = 0.019). Also, the species *Candidatus Heimdallarchaeota archaeon* was significantly higher at the shallower Station A (p = 0.019) (Fig 4a–4c). The differences in abundance of archaeal communities versus bacterial communities were less pronounced between stations K6 and 18 (p = 0.9) and K6 and A (p = 0.07). Among the bacterial families, Pelagibacteriaceae (0.00004), Rhodobacteracea (0.001), and Synechoccaceae (p = 0.0003) were significantly different in abundances at station 18 versus A (S11 File). The families Rhodobacteraceae (p = 0.0005), Synechoccaceae (p = 0.001), and Erythrobacteraceae (p = 0.016) were significantly differentially abundant between stations K6 and 18 (S12 File). The families Pelagibacteraceae (p = 0.00003) and Erythrobacteraceae (p = 0.02) were significantly differentially abundant between stations K6 and A (S13 File). Abundances of unclassified Pelagibacteriaceae (p = 0.00004), unclassified Rhodobacteraceae (p = 0.001), and *Synechococcus* (p = 0.0003) were also significantly different between stations 18 and A. The latter two were also significantly different between stations K6 and 18. The unclassified Pelagibacteriaceae (p = 0.00003), Altererythrobacter (p = 0.02) and Marinobacter (0.04) were significantly different between stations K6 and A. Differences in the abundances of further unclassified species of the same families also depicted p values less than 0.05 indicative of spatial variations between stations 18 and A, K6 and 18, and K6 and A.

Significant differences at the family, genus, and species levels between the three stations were observed (Q value < 0.05). Comparisons between Stations 18 and A revealed that 6 families (Aeromonadaceae (q = 0.013), Bacteroidaceae (q = 0.013), Prevotellaceae (q = 0.013), Sinobacteraceae (q = 0.013), Pelagibacteraceae (q = 0.02), and Flavobacteriaceae (q = 0.23)) showed

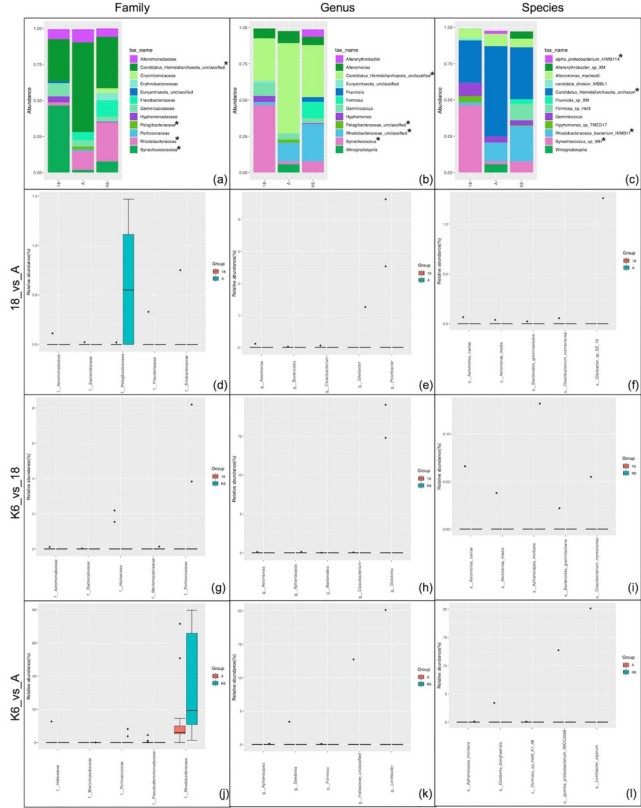

**Fig 4. Comparison of taxonomic distribution of bacterial and archaeal populations relative to sampling location (Station 18 (n = 12) versus Station A (n = 12) versus Station K6 (n = 12)).** (a-c) Results are depicted at the (a) family, (b) genus, and (c) species levels. The relative abundance of the twelve most abundant taxonomic groups per classification are depicted. *p<0.02 (Wilcoxon matched-pairs signed rank/Mann-Whitney test). (d-l) Results are depicted in the form of differential abundance analysis at the (d,g,j) family, (e,h,k) genus, and (f,i,l) family levels. The top differentially abundant taxa are displayed. All displayed taxa have p value <0.001 (non-parametric t-test).

significantly differential abundance between the two stations (Fig 4d and S14 File). In total, 10 genera showed statistically significant differential abundance (q<0.008) between Stations 18 and A. The top six differentially abundant genera were *Aeromonas* (q = 0.008), *Bacteroides* (q = 0.008), *Cloacibacterium* (q = 0.008), *Gilvibacter* (q = 0.008), and *Polaribacter* (q = 0.008) (Fig 4e and S15 File). Overall, 20 species showed significantly preferential abundance between Stations 18 and A, 15 of which showed significantly greater abundance at Station A, and 5 of which showed significant preferential abundance at Station 18. Top six differentially abundant species were *Aeromonas caviae*, *Aeromonas media*, *Bacteroides graminisolvens*, *Cloacibacterium normanense*, and *Gilvibacter* sp SZ19 (q<0.05) (Fig 4f and S16 File).

Comparisons between Stations K6 and 18 revealed that 9 families showed significant differential abundance between the two stations. 6 families showed significant preferential abundance at Station 18 (*Aeromonadaceae*, *Bacteroidaceae*, *Halieaceae*, *Prevotellaceae*, *Pseudoalteromonadaceae*, and *Sinobacteraceae*) while 3 families showed significant preferential abundance at Station K6 (*Merismopediaceae*, *Porticoccaceae*, and *Pelagibacteraceae*). At the species level, more than 50% (26/50) species showed significant differential abundance between Stations K6 and 18. 22 species showed significant preferential abundance at Station18, while only 4 species showed significant preferential abundance at Station K6. The top six

differentially abundant families, genera, and species at FDR (q) <0.05 are presented on Fig 4g–4i and S17–S19 Files.

Finally, comparisons between Stations K6 and A showed that only 4 of the 27 families showed significantly differential abundance between the two stations. 2 families (*Halieaceae* and *Pseudoalteromonadaceae* (q = 0.02)) showed preferential abundance at Station A, and 2 families (*Merismopediaceae* and *Porticoccaceae* (q = 0.02)) showed preferential abundance at Station K6. At the species level, 15/50 species showed significant differential abundance between the two stations with 10 species showing significant preferential abundance at Station A and only 5 showing significant preferential abundance at Station K6. Only the first six taxa at the family (Fig 4j), genus (Fig 4k) and species (Fig 4l) level are represented on the box plots (S20–S22 Files).

## Seasonal variations in community profiles

Season (autumn/winter) as an experimental factor proved to be a greater discriminating factor affecting relative microbial abundance compared to spatial distribution and water column depth. The effects were observed upon assessment of relative abundance (significance at p<0.01) of both archaeal and bacterial populations (Fig 5a–5c and S23 File). The archaeal population showed a significant increase (log2 median ratio of relative abundance between autumn and winter = 2.6) while the bacterial population showed a significant decrease in the winter season (log2 median ratio = -1.29). The archaeal phyla Candidatus_Heimdallarchaeota displayed significant variations in abundance between the two seasons at p = 0.01. Among the bacterial phyla Bacteroidetes and Cyanobacteria displayed significantly differential abundance at p = 0.009 and 0.006, respectively. At class level specifically, members of Alphaproteobacteria showed preferential abundance during the winter season (p = 0.006) while members of the class Gammaproteobacteria showed preferential abundance during the autumn season (p = 0.003). Closer examination of members of the class Alphaproteobacteria at deeper taxonomic levels revealed that members of the order Rhodobacterales (family Rhodobacteraceae,

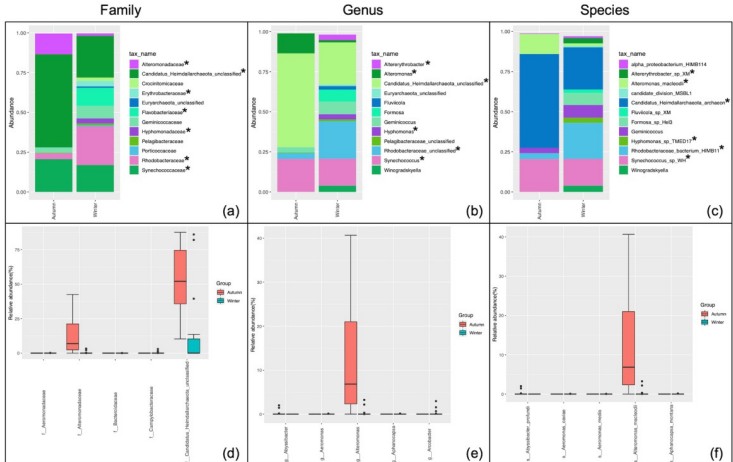

**Fig 5. Comparison of taxonomic distribution of bacterial and archaeal populations relative to sampling season (autumn (September–November; n = 18) versus winter (December–February; n = 18)).** (a-c) Results are depicted at the (a) family, (b) genus, and (c) species levels. The relative abundance of the twelve most abundant taxonomic groups per classification are depicted. *p<0.04 (Wilcoxon matched-pairs signed rank/Mann-Whitney test). (d-f) Results are depicted in the form of differential abundance analysis at the (d) family, (e) genus, and (f) species levels. All displayed taxa have q-value≤0.002; p-value<0.001 (non-parametric t-test).

genus Rhodobacteraceae_unclassified, and species *Rhodobacteraceae_bacterium_HIMB11;* family Hyphomonadaceae, genus *Hyphomonas*, and species *Hyphomonas_sp_TMED17*) exhibited significantly higher abundance during the winter as did one member of the order Sphingomonadales (family Erythrobacteraceae, genus *Altererythrobacter*, and species *Altererythrobacter_sp_XM_24bin4*). Within the class Gammaproteobacteria, members of the order Alteromonadales (family Alteromonadaceae, genera *Alteromonas* and *Marinobacter*, and species *Alteromonas_macleodii* and *Marinobacter_hydrocarbonoclasticus*) showed significantly preferential abundance during the autumn season (p<0.002 for the order).

Both detectable classes within the phyla Cyanobacteria and Bacteroidetes (Cyanobacteria_unclassified and Flavobacteriia) showed significantly different abundance between the autumn and winter seasons. Specifically, the class Cyanobacteria_unclassified showed significantly preferential abundance during the autumn season (p = 0.006) while the class Flavobacteriia showed preferential abundance during the winter season (p = 0.009). Specifically, members of the class Cyanobacteria (order Synechococcales, family Synechococcaceae, genus *Synechococcus*, and species *Synechococcus_sp_WH_8109*) showed significantly preferential abundance during the autumn season (p = 0.006).

Finally, members of the class Flavobacteriia (order Flavobacteriales, and family Flavobacteriaceae) showed a statistically significant preference for abundance during the winter season (p<0.04).

Differential abundance analysis was also performed to investigate statistically significant differences in abundance with respect to seasonality (Fig 5d–5f and S24–S26 Files). Results show 15 families (11 abundant during winter and 4 in autumn, q<0.01), showed statistically significant differential abundance with respect to season. Indicator families in winter were Aeromonadaceae, Halieaceae, Porticoccaceae, Oceanospirillaceae, Camplyobacteraceae, Rhodobacteraceae, Bacteroidaceae, Prevotellaceae, Flavobacteriaceae, Merismopediaceae and Microbacteriaceae. Those more common in autumn (q = 0.002) were unclassified Candidatus Heimdallarchaeota, Alteromonadaceae, Salinisphaeraceae, and Sinobacteraceae (S24 File).

Further probing revealed the genera within each of these families that were significantly preferentially abundant during each of the two seasons. Of the 39 genera detected, 25 showed significant differential abundance in one season versus the other (19 genera with significant preferential abundance during winter and 6 with significant preferential abundance during autumn) (q = 0.0016) (S25 File).

Specifically, the genera that were more abundant during winter were *Aeromonas*, *Bermanella*, unclassified *Halieaceae*, *Porticoccus*, *Lentibacter*, *Nereida*, *Phaeobacter*, *Thalassobacter*, *Arcobacter*, *Cloacibacterium*, *Dokdonia*, *Formosa*, *Gilvibacter*, *Polaribacter*, *Winogradskyella*, *Prevotella*, *Candidatus Aquiluna*, *Aphanocapsa*. The genera that were more significantly plentiful during autumn included unclassified *Candidatus_Heimdallarchaeota*, *Rhodovulum*, *Abyssibacter*, *Alteromonas*, *Marinobacter*.

Of the 50 species tested, 35 showed significant differential abundance between autumn and winter (24 were more significantly abundant during winter and 11 were more significantly abundant during autumn) (S26 File).

The species that showed preferential abundance in the winter were *Candidatus_Aquiluna_sp_XM*, *Bacteroides_graminisolvens*, *Prevotella_copri*, *Cloacibacterium_normanense*, *Dokdonia_donghaensis*, *Formosa_sp_Hel3_A1_48*, *Gilvibacter_sp_SZ_19*, *Polaribacter_sp_MED152*, *Winogradskyella_sp*, *Aphanocapsa_montana*, *Lentibacter_algarum*, *Nereida_ignava*, *Phaeobacter_italicus*, *Rhodobacteraceae_bacterium_SB2*, *Thalassobacter_stenotrophicus*, *Arcobacter_cryaerophilus*, *Aeromonas_caviae*, *Aeromonas_media*, *Aeromonas_veronii*, *Pseudoalteromonas_marina*, *Pseudoalteromonas_tetraodonis*, *gamma_proteobacterium_IMCC3088*, *Porticoccus_sp*, and *Bermanella_sp*.

Species that showed significant preferential abundance during autumn, were unclassified *Candidatus_Heimdallarchaeota_archaeon*, *Synechococcus_sp_KORDI_52*, *Synechococcus_sp_RCC307*, *Synechococcus_sp_RS9916*, *Rhodovulum_sp*, *Alteromonas macleeodii*, *Pseudoalteromonas ruthenica*, *Marinobacter adhaerens Marinobacter hydrocarbonoclasticus*, *Sinimarinibacterium flocculans*, *Abyssibacter_profundi*.

## Alpha and beta diversity analyses

Alpha diversity analysis revealed interesting trends in diversity patterns with respect to depth (Fig 6a), season (Fig 6b), and location (Fig 6c). Namely, Chao diversity index analysis reveals significantly greater diversity in surface samples versus bottom samples (p = 0.04) as well as significantly greater diversity in autumn samples versus winter samples (p = 0.01). Results from other alpha diversity metrics including Shannon, Simpson, and Inverse Simpson indices yielded largely consistent trends in diversity with respect to depth and seasonality (S4 Table).

Alpha diversity analysis with respect to sampling location (Station 18 versus Station A versus Station K6) revealed no significant differences in diversity between the three stations (p>0.1). However, Station A, located northeast of Kuwait Bay in a narrow waterway between

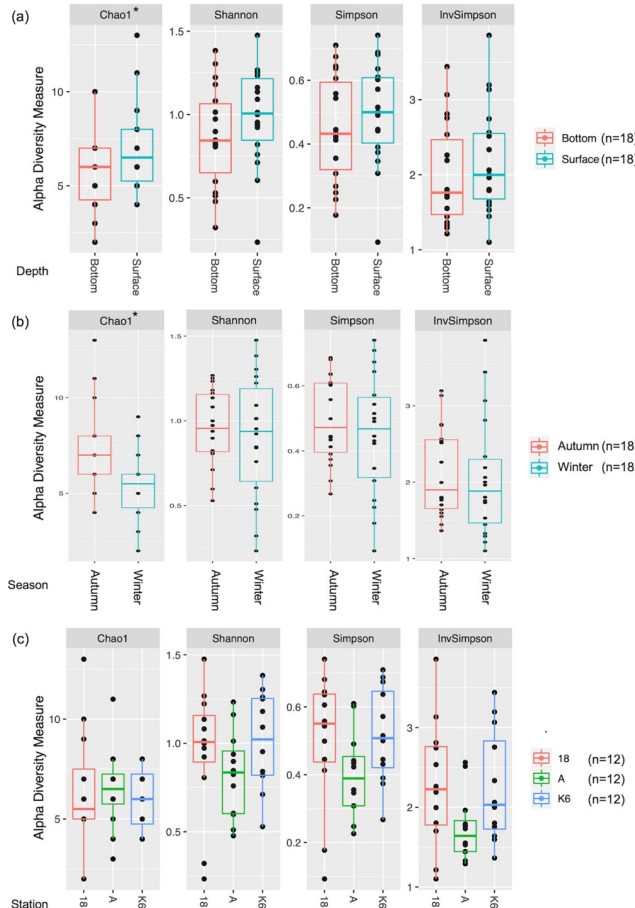

**Fig 6. Alpha diversity metrics assessed with respect to depth, seasonal variation, and spatial distribution.** Alpha diversity metrics of bacterial and archaeal communities in (a) surface (n = 18) versus bottom (n = 18) samples, (b) autumn (n = 18) versus winter (n = 18) seasons, and (c) Station 18 (n = 12) versus Station A (n = 12) versus Station K6 (n = 12). Boxplots depict the 25th percentile, median, and 75th percentile. *p<0.05 (ANOVA).

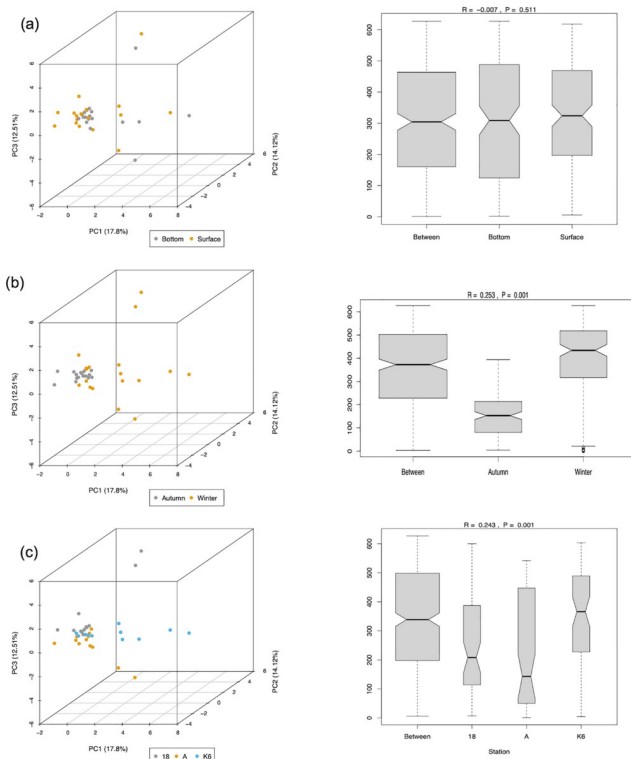

**Fig 7.** Beta diversity clustering assessed based on principal component analysis (PCA) and Analysis of Similarity (ANOSIM) of bacterial genera between (a) surface and bottom samples, (b) autumn and winter samples, and (c) samples from Stations 18, A, and K6. 3D scatter plots show principal components PC1, PC2, and PC3. Proportion of variance for PC1, PC2, and PC3 are shown.

the Kuwaiti coastline and the Kuwaiti island of Bubiyan consistently displayed less diversity compared to Stations 18 and K6, which largely showed similar diversity.

Finally, principal component analysis (PCA) and Analysis of Similarity (ANOSIM) were performed to assess beta diversity clustering patterns with respect to depth (Fig 7A), season (Fig 7B), and location (Fig 7C) (S27–S29 Files). Results showed that while the surface and bottom microbial communities did display partially overlapping clusters, there were no significant differences in beta diversity clustering with respect to depth (p = 0.5). Interestingly, however, results revealed significantly increased beta diversity in the winter versus autumn seasons (p = 0.001). Furthermore, PCA revealed significant differences in beta diversity clustering between the three sampling stations. The three stations did contain overlapping clusters, but the three stations also contained some outlying clusters that were significantly different between the three stations (p = 0.001).

## Physicochemical characteristics of the seawater and surrounding environment at the time of sampling

Fluctuation patterns in seawater temperature, dissolved oxygen level, and salinity level measurements obtained from both the surface and the bottom of the water column from each of the three sampling locations during the six-month sampling period were assessed. The average recorded seawater temperatures were 28.70 ˚C and 17.95 ˚C for the autumn and winter seasons, respectively, the average recorded dissolved oxygen levels were 5.74 ml/L and 7.67 ml/L

for the autumn and winter seasons, respectively, and the average recorded salinity levels were 41.73 psu and 39.69 psu for the autumn and winter seasons, respectively (S1a–S1c and S2 Figs and S2 Table). Average sample acquisition depth ranged from 1.29 m to 11.54 m for the surface and bottom samples, respectively (S1e Fig and S2 Table).

Measured levels of dissolved oxygen show similar patterns in all three stations. Namely, apart from a dip during the month of October, particularly in the bottom samples, dissolved oxygen levels showed a steady increase throughout the six-month study period. This pattern is in direct contrast with the pattern of seawater temperature, which unsurprisingly shows a steady decrease throughout the six-month study period. Surface and bottom measurements from each of the three stations for all three physicochemical parameters assessed were largely congruent (S2 Fig), with Station A measurements showing the largest degree of overlap between surface and bottom readings likely due to its shallow nature (S2 Table).

### Analysis of correlation between community profiles and metadata

Kendall and Spearman methods were used to assess whether the correlation between abundance of microbial species and seawater metadata including seawater temperature, salinity, and dissolved $O_2$ levels. Results revealed that the abundance of 5 out of 30 species displayed significant correlation with seawater temperature as assessed by both the Kendall ($p<0.04$) and Spearman methods ($p<0.04$) (S30–S32 Files). The species whose abundance was significantly correlated with seawater temperature were *alpha_proteobacterium_HIMB114*, *Altererythrobacter_sp_XM_24bin4*, *Candidatus_Heimdallarchaeota_archaeon*, *Formosa_sp_Hel3_A1_48*, and *Synechococcus_sp_RS9916*.

Results also revealed that the abundance of 5 out of 30 species showed significant correlation with salinity levels as assessed by both the Kendall ($p<0.035$) and Spearman (0.04) methods. The species whose abundance significantly correlated with salinity levels were *alpha_proteobacterium_HIMB114*, *Candidatus_Heimdallarchaeota_archaeon*, *Altererythrobacter_sp_XM_24bin4*, *Fluviicola_sp_XM_24bin1*, and *Formosa_sp_Hel3_A1_48*.

Finally, results revealed that 6 out of 30 species showed significant correlation with dissolved $O_2$ levels as assessed by both the Kendall ($p<0.03$) and Spearman ($p<0.02$) methods. Namely, abundance of the species *alpha_proteobacterium_HIMB114*, *Altererythrobacter_sp_XM_24bin4*, *Formosa_sp_Hel3_A1_48*, *Pseudoalteromonas_ruthenica*, *Candidatus_Heimdallarchaeota_archaeon*, and *Synechococcus_sp_RS9916* was significantly correlated with dissolved $O_2$ levels.

Overall, correlation between species abundance and the various physicochemical parameters described above does not appear to be restricted to any specific domain or phylum as both members of the archaeal and bacterial domains showed significant correlations with the three physicochemical parameters assessed, and members of various bacterial phyla including *Alphaproteobacteria*, *Gammaproteobacteria*, *Cyanobacteria*, and *Bacteroidetes* all showed significant variations in abundance relative to the physicochemical parameters studied.

## Discussion

Despite the demonstrated significance of shotgun metagenomic approaches, there have been very limited studies in the past involving community-level profiling of microbial communities within the Kuwaiti environment. Previous studies on the marine environment in Kuwait involved the use of DNA fingerprinting to examine the spatiotemporal diversity of bacterioplankton [42], sequencing of 16S rRNA to identify resident bacteria [43, 44], sequencing of 18S rRNA to assess distribution and diversity of eukaryotic microalgae [45] and resistome profiling of polluted marine sediments [46]. Importantly, marine bioprospecting efforts in the

Kuwaiti marine environment have led to the discovery of marine bacteria that produce poly-unsaturated fatty acids, which are essential nutrients to vertebrates and have proven beneficial effects against various ailments [47]. The utility of such discoveries would be significantly expanded with the presence of shotgun metagenomic data that could be probed for identification of functional genetic components underlying such desirable traits. Besides the marine environment, metagenomics approaches have been employed in Kuwait to develop comprehensive profiles of bacterial and fungal communities associated with aerosols in urban versus rural areas [14], hospitals [48], as well as to analyze viral diversity in respiratory samples from patients with respiratory tract infections [49].

## Taxonomic diversity in Kuwaiti territorial waters

In the present study, the presence of both archaeal and bacterial communities in seawater samples collected from the surface and bottom of the water column at three different stations located in and around the unique shallow water body of Kuwait Bay in the northwest region of the Arabian Gulf monthly over a six-month period spanning two seasons (autumn (September-November) and winter (December-February)) was recorded.

Planktonic bacteria and archaea co-exist and play an important role in ecotype diversification [50], determining the biogeographic patterns of coastal wetlands [51], and global carbon cycling [52]. Archaeal communities had a RA of 42.9% while bacteria exhibited a RA of 57.1%. Similar observations were recorded in the coastal wetlands of China [51]. However, only 14.3% of reads classified as Archaea in the oceanic water columns of Monterey Bay [50]. The most prominent bacterial phylum in the study samples was Proteobacteria (55.4%), which was followed in abundance by the phyla Cyanobacteria (32.8%) and Bacteroidetes (11.8%). These results were in partial agreement with the bacterial phyla observed in the anoxic zone of the Cariaco basin [53]. The archaeal phyla of Euryarchaeota along with Crenarchaeota, rather than Candidatus_Heimdallarchaeota as reported in this study, were found in the water masses of the Tyrrhenian Sea [54]. Variations in predominant bacterial and archaeal communities are expected due to the differences in geographical boundaries as well as variations in the technical approaches followed during each investigation.

The present assumptions pertaining to the dominant species are made on the basis of environmental DNA (eDNA) sequencing data. Methodologically, eDNA sequencing is limited in differentiating between live and dead microbes. Future investigations on living bacterial and archaeal assemblages using a metatranscriptomics approach are therefore recommended.

## Archaeal and bacterial genera in Kuwaiti waters

The bacterial phyla detected within the study samples were distributed into a number of classes, orders, families, genera, and species, whereas the subordinate archaeal taxons were predominated by members of Candidatus_Heimdallarchaeota_unclassified. The lack of detailed taxonomic profiling beyond the phylum level may be attributed to the limited resources available on archaeal biology [52]. Within the bacterial domain, the predominance of members of Rhodobacteraceae within the study samples is most likely due to the presence of organic and inorganic compounds, sulfur oxidation, aerobic anoxygenic photosynthesis, carbon monoxide oxidation, and the production of secondary metabolites in the oceanic waters of Kuwait [55]. Additionally, the average combined nitrate and nitrite concentrations measured between November and December 2018 at these stations ranged between 0.88–9.82 μM [31]. In another study, the average nitrate concentration was reported at 1.15 mg/L during the summer and 10.9 mg/L during the winter [56]. Such concentrations lead to the generation of a hypoxic environment hence the dominance of archaeal communities. Furthermore, the presence of

members of Rhodobacteraceae has also been linked with algal biofilms, which are commonly found in Kuwaiti waters [45, 57]. This is further corroborated by the presence of *Synechococcus*, a genus of cyanobacterial origin [58]. The archaeal genus *Heimdallarchaeota* is more closely related to eukaryotes, and may also rely on the exchange of hydrogen, electrons, and/or simple carbon compounds for nutrition [52]. Future explorations into metabolic activities, with a focus on nitrate and sulphate metabolism within these waters will assist in correlating the taxonomic and functional profiles.

## Variations in relative abundances of bacterial and archaeal communities with respect to depth

Differences in microbial assemblages and abundances were compared between samples collected from the surface versus bottom of the water column (Fig 3 and S7–S10 Files). Bacterial and archaeal assemblages were reported to increase from middle to high latitudes in deep sea-ecosystems of the Arctic Ocean [59]. Presently we report 6, 9, and 18 bacterial families, genera, and species, respectively that differ in abundance (q < 0.05) according to depth. This is attributed towards their sensitivity to changes in temperature and nutrient availability [59]. Surprisingly, none of the archaeal taxa detected within the study samples showed variations with respect to depth down the water column. Kuwait bay is a shallow water body with an average depth of 5.2 m and a tidal range of 4 m [60]. The profile reported in the present investigation is at an altitudinal difference of 10 m, as compared to 200–500 m in Monterey Bay [50] and 400–5570 m in the Arctic Ocean [59]. Our results were corroborated by the alpha and beta diversity analyses performed as none of the alpha diversity indices measured (Shannon, Simpson, or Inv Simpson) except Chao1 showed significance at a confidence interval of 99.95% and an ANOSIM of $r^2$ = -0.007, (p = 0.511) between the two depths indicating negligible diversity with respect to depth down the water column (S4 Table and S29 File).

## Spatial variations in taxonomies

Spatial variations are common among microbial communities [61–63]. This is most likely due to the presence of environmental gradients and variations in physicochemical parameters between sampling location. Although variations in differential abundance of key genera were detected at each sampling location, measurements of species richness and evenness revealed negligible variation (ANOVA p<0.05). Furthermore, a moderately defined community structure (ANOSIM $r^2$ = 0.243; p>0.001) was also noted (S29 File). The sampling sites for the present study all lie within a semi-enclosed stretch of 720 km$^2$, which likely accounts for the limited variability [60]. Differences in dominance of archaeal and bacterial genera is likely due to the various anthropogenic influences exerted at each location. For instance, Station K6, which is located within Kuwait Bay, receives sewage discharge as well as discharge from desalination plants and industrial sources. Meanwhile, station A is located northeast of the Bay between the Kuwaiti coastline and the Kuwaiti island of Bubiyan, and Station 18, located southeast of the Bay, is the most remote and thus the least subject to anthropogenic activities (Fig 4 and S11–S22 Files). Limited spatial variations in microbial communities were also reported in the atmospheric environments of Kuwait [14, 48, 64, 65]. Plant genetic diversity within the country is also limited owing to the narrow geographic extent [14, 66–68].

## Temporal variations in bacterial and archaeal communities

Microbes have inherent abilities to adapt according to changing environmental conditions. Seasonal variations in microbial community composition have been reported in thermohaline waters of the Antarctic [61], benthic deep-sea waters of the Atlantic Ocean, Labrador Sea,

Subarctic waters, and Mediterranean waters [59]. Results from this study are in agreement with the aforementioned studies and it is likely that temporal fluctuations between the autumn and winter seasons were the key drivers of the differential abundances observed in both bacterial and archaeal populations. Abundance of members of the archaeal population showed a significant increase ($\log_2$ median ratio of RA between autumn and winter = 2.6) while the bacterial population showed a significant decrease in the winter season ($\log_2$ median ratio = -1.29) (Fig 5 and S23–S26 Files). These findings were supported by analyses of alpha and beta diversity as well (ANOSIM $r^2$ = 0.253; p>0.01) (S29 File). Average seawater temperature, dissolved oxygen levels, and salinity levels all differed significantly between the two seasons. It is prudent to study the community composition in summer and spring seasons at a higher sampling frequency.

## The effects of physicochemical conditions on bacterial and archaeal communities

Analysis of various physicochemical parameters revealed significant differences in temperature, salinity, and dissolved oxygen levels throughout the study period. Specifically, there was a 10.75˚C drop in average seawater temperature, a 1.93 ml/L increase in average dissolved oxygen levels, and a 2.04 psu drop in average salinity levels between the autumn and winter seasons (S1a–S1c Fig and S2 Table). Investigations into the effects of these variations on community composition revealed that the abundance of 5 species (*alpha_proteobacterium_HIMB114*, *Altererythrobacter_sp_XM_24bin4*, *Candidatus_Heimdallarchaeota_archaeon*, *Formosa_sp_Hel3_A1_48*, and *Synechococcus_sp_RS9916)* displayed significant correlation with seawater temperature as assessed by both the Kendall (p<0.04) and Spearman methods (p<0.04) (S30–S32 Files). Furthermore, 6 species showed significant correlation with dissolved $O_2$ levels as assessed by both the Kendall (p<0.03) and Spearman (p<0.02) methods.

## Potential metabolic functions carried out by the identified microbes

The identified species are anticipated to mediate several metabolic processes. Of note, the relative abundance of the species, and by extension the metabolic processes they carry out, are subject to change with respect to environmental variations during the sampling period. The pathways of energy metabolism, lipid metabolism, DNA repair and transcription regulation, which were reported elsewhere to be connected to the overall stress response, are likely to be active given the stress inevitably induced by the extreme environmental conditions [46]. Importantly, the presence of pollutants such as microplastics in the marine environment also introduces stress on the bacterial population and has been demonstrated to activate additional processes involved in energy conversion, carbohydrate metabolism, and transport metabolism [26, 69–71]. Future studies investigating the functional profiles of these bacterial and archaeal communities will formulate a solid baseline describing biogeochemical and other functional processes that are active in the marine waters of Kuwait.

## Conclusions

Overall, the present study is the first of its kind to investigate bacterial community structure using shotgun metagenomics in Kuwaiti territorial waters of the northwest Arabian Gulf. Furthermore, this study includes the first documented investigation of archaeal community structure within this unique water body. The findings show compelling patterns in spatiotemporal variability with respect to factors such as depth, season, and location. These findings are critical to laying the foundation for future research aimed at delving deeper into the microbial

community structures within this distinct marine environment. Moreover, the results will be useful for examining the functional activities these microbial populations employ to survive and thrive in the region's harsh conditions.

## Supporting information

**S1 Fig. Analysis of average levels of various parameters (salinity, dissolved oxygen, temperature, and depth) measured at the sampling sites throughout the study period.** (a-c) Variations in average levels of measured physicochemical parameters between autumn and winter. Average values of seawater (a) temperature, (b) dissolved oxygen levels, and (c) salinity levels were compared between the autumn (n = 18) and winter (n = 18) seasons. Error bars represent standard deviations. $^*p<1E-4$, $^{**}p<1–8$, $^{***}p<1E-11$ (Student's t-test). (de) Station depth and sample acquisition depth. (d) Total station depth measured during each of the six months of sampling for each of the three stations (K6, 18, and A). (e) Average depth of sample acquisition for surface (n = 18) and bottom (n = 18) samples. Error bars represent standard deviations. $^*p<1–7$ (Student's t-test).
(PNG)

**S2 Fig. Temporal fluctuations in measured physicochemical parameters of the seawater at both the surface and bottom of each of the three sampling locations.** Values of seawater (a) temperature, (b) dissolved oxygen levels, and (c) salinity levels were measured at the time of sampling at both the surface (blue line) and bottom (red line) of each of the three sampling stations (Station 18 (left), Station A (middle), and Station K6 (right)). Values for each measured parameter are depicted for each month of the six-month sampling period (September 2019–February 2020).
(PNG)

**S1 Table. Log of sampling location, date, and time for the six-month sampling period.**
(PNG)

**S2 Table. Physicochemical characteristics of the seawater at the time of sampling from the three sampling stations, 18, A, and K6.**
(PNG)

**S3 Table. Physicochemical characteristics of the surrounding environment at the time of sampling from the three sampling stations, 18, A, and K6.**
(PNG)

**S4 Table. Alpha diversity analysis detailed metrics and analysis of variance (ANOVA) p-values.**
(PNG)

**S1 File. Relative abundance levels used to generate barplots of bacterial and archaeal community structures in all project samples (family-level analysis).**
(XLSX)

**S2 File. Relative abundance levels used to generate barplots of bacterial and archaeal community structures in all project samples (genus-level analysis).**
(XLSX)

**S3 File. Relative abundance levels used to generate barplots of bacterial and archaeal community structures in all project samples (species-level analysis).**
(XLSX)

**S4 File. Core microbiome analysis (family-level).**
(XLSX)

**S5 File. Core microbiome analysis (genus-level).**
(XLSX)

**S6 File. Core microbiome analysis (species-level).**
(XLSX)

**S7 File. Differential heat tree analysis—Surface versus bottom samples.**
(XLSX)

**S8 File. Differential abundance analysis via Metastats—Surface versus bottom samples—Family-level analysis.**
(XLSX)

**S9 File. Differential abundance analysis via Metastats—Surface versus bottom samples—Genus-level analysis.**
(XLSX)

**S10 File. Differential abundance analysis via Metastats—Surface versus bottom samples—Species-level analysis.**
(XLSX)

**S11 File. Differential heat tree analysis—Station 18 versus Station A.**
(XLSX)

**S12 File. Differential heat tree analysis—Station K6 versus Station 18.**
(XLSX)

**S13 File. Differential heat tree analysis—Station K6 versus Station A.**
(XLSX)

**S14 File. Differential abundance analysis via Metastats—Station 18 versus Station A—Family-level analysis.**
(XLSX)

**S15 File. Differential abundance analysis via Metastats—Station 18 versus Station A—Genus-level analysis.**
(XLSX)

**S16 File. Differential abundance analysis via Metastats—Station 18 versus Station A—Species-level analysis.**
(XLSX)

**S17 File. Differential abundance analysis via Metastats—Station K6 versus Station 18—Family-level analysis.**
(XLSX)

**S18 File. Differential abundance analysis via Metastats—Station K6 versus Station 18—Genus-level analysis.**
(XLSX)

**S19 File. Differential abundance analysis via Metastats—Station K6 versus Station 18—Species-level analysis.**
(XLSX)

**S20 File. Differential abundance analysis via Metastats—Station K6 versus Station A—Family-level analysis.**
(XLSX)

**S21 File. Differential abundance analysis via Metastats—Station K6 versus Station A—Genus-level analysis.**
(XLSX)

**S22 File. Differential abundance analysis via Metastats—Station K6 versus Station A—Species-level analysis.**
(XLSX)

**S23 File. Differential heat tree analysis—Autumn versus winter seasons.**
(XLSX)

**S24 File. Differential abundance analysis via Metastats—Autumn versus winter seasons—Family-level analysis.**
(XLSX)

**S25 File. Differential abundance analysis via Metastats—Station K6 versus Station A—Genus-level analysis.**
(XLSX)

**S26 File. Differential abundance analysis via Metastats—Station K6 versus Station A—Species-level analysis.**
(XLSX)

**S27 File. Summary of Principal Component Analysis (PCA) findings.**
(XLSX)

**S28 File. Detailed Principal Component Analysis (PCA) findings.**
(XLSX)

**S29 File. Analysis of Similarity (ANOSIM) findings.**
(TXT)

**S30 File. Analysis of correlation between physicochemical parameter measurements and OTU abundance (family-level)—Results from both Kendall and Spearman methods.**
(XLSX)

**S31 File. Analysis of correlation between physicochemical parameter measurements and OTU abundance (genus-level)—Results from both Kendall and Spearman methods.**
(XLSX)

**S32 File. Analysis of correlation between physicochemical parameter measurements and OTU abundance (species-level)—Results from both Kendall and Spearman methods.**
(XLSX)

## Acknowledgments

The authors would like to thank Mr. Sasi Kumar Chellakan and Mr. Sainulabdeen Shajim for their assistance during the seawater sampling expeditions and Dr. Faiza Al-Yamani for her revision of the manuscript. The authors would also like to acknowledge Dr. Fathima Thaslim and Mr. Imtiaz Ahmed for administrative assistance.

## Author Contributions

**Conceptualization:** Saja A. Fakhraldeen, Surendraraj Alagarsamy.

**Data curation:** Saja A. Fakhraldeen, Sakinah Al-Haddad, Abdulmuhsen K. Ali.

**Formal analysis:** Saja A. Fakhraldeen, Nazima Habibi.

**Funding acquisition:** Saja A. Fakhraldeen.

**Investigation:** Saja A. Fakhraldeen, Sakinah Al-Haddad, Surendraraj Alagarsamy.

**Methodology:** Saja A. Fakhraldeen, Sakinah Al-Haddad, Abdulmuhsen K. Ali.

**Project administration:** Saja A. Fakhraldeen.

**Resources:** Saja A. Fakhraldeen, Surendraraj Alagarsamy, Sabeena F. K. Habeebullah, Walid M. Al-Zakri.

**Software:** Saja A. Fakhraldeen, Nazima Habibi, Abdulmuhsen K. Ali.

**Supervision:** Saja A. Fakhraldeen.

**Validation:** Saja A. Fakhraldeen, Sakinah Al-Haddad.

**Visualization:** Saja A. Fakhraldeen, Nazima Habibi, Abdulmuhsen K. Ali.

**Writing – original draft:** Saja A. Fakhraldeen, Nazima Habibi.

**Writing – review & editing:** Saja A. Fakhraldeen, Nazima Habibi, Surendraraj Alagarsamy, Sabeena F. K. Habeebullah.

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
