## [Decision Letter · Decision Letter 0]

4 Jun 2023

PONE-D-23-05247Diversity and spatiotemporal variations in bacterial and archaeal communities within Kuwaiti territorial waters of the Northwest Arabian GulfPLOS ONE

Dear Dr. Fakhraldeen,

Thank you for submitting your manuscript to PLOS ONE. After careful consideration, we feel that it has merit but does not fully meet PLOS ONE’s publication criteria as it currently stands. Therefore, we invite you to submit a revised version of the manuscript that addresses the points raised during the review process.

My opinion is kind of inclined with reviewer #2 that the description of the methods are not complete and that raised serious concerns by the reviewers. In that case a rigorous revision and careful rewrite is recommended for the manuscript as per reviewers' suggestion. Secondly, there are missing parts in the findings/analysis that might be potentially help conclude the overall story. Although the uniqueness of the study has been greatly appreciated, I would strongly recommend to follow a major revision that makes this article upto the version it should be published in.

We look forward to receiving your revised manuscript.

Kind regards,

Tarunendu Mapder, Ph.D.

Academic Editor

PLOS ONE

Journal Requirements:

Reviewers' comments:

Reviewer's Responses to Questions

**Comments to the Author**

1. Is the manuscript technically sound, and do the data support the conclusions?

Reviewer #1: Yes

Reviewer #2: Partly

2. Has the statistical analysis been performed appropriately and rigorously? 

Reviewer #1: Yes

Reviewer #2: No

3. Have the authors made all data underlying the findings in their manuscript fully available?

Reviewer #1: Yes

Reviewer #2: Yes

4. Is the manuscript presented in an intelligible fashion and written in standard English?

Reviewer #1: Yes

Reviewer #2: Yes

5. Review Comments to the Author

Reviewer #1: This study analyzes the bacterial and archaeal community structures present in the unique waters of Kuwait, located in the northwest Arabian Gulf. The study aims to identify resident organisms and understand the dynamic nature of their existence in an environment prone to various environmental and anthropogenic stressors. The study utilizes a shotgun metagenomics approach and analyzes variations in community structures with respect to depth, season, and location, as well as their susceptibility to changes in abundance with respect to various physicochemical parameters. The results indicate an approximately even abundance of archaeal and bacterial communities but significantly greater diversity among the bacterial population. The study provides a framework for future investigations of functional genetic adaptations developed by resident biota attempting to survive in the extreme conditions of the unique aquatic ecosystem.

Line 271: Since Rhodobacteraceae_unclassified (14.41%) does not indicate genus name, it could be avoided from the genera level result descriptions.

Line 272: Similarly Candidatus_Heimdallarchaeota unclassified (13.10%),

Pelagibacteraceae unclassified, Euryarchaeota unclassified (4.37%) could be avoided.

Line 275-281: Since the usage of short gun sequencing is to highlight the species richness, the results of species level resolution could be highlighted better: example; Candidatus Heimdallarchaeota archaeon, Alteromonas macleodii, Marinobacter hydrocarbonoclasticus,

Line 665 – 673: Conclusion should be re-written.

Example:

In summary, the current study is groundbreaking as it is the first to use shotgun metagenomics to explore bacterial community structure and to investigate archaeal community structure in Kuwaiti territorial waters of the northwest Arabian Gulf. The findings show compelling patterns in spatiotemporal variability concerning factors such as depth, season, and location. These findings are critical to laying the foundation for future research to delve deeper into the microbial community structures within this distinct marine environment. Moreover, the results will be useful for examining the functional activities these microbial populations undertake to survive and thrive in the region's harsh conditions.

Reviewer #2: The authors studied the microbial diversity of Kuwaiti territorial waters of Arabian sea by shot gun metagenomics approach at three sampling points for six months. This study is certainly appreciable for its unique sampling site. There are some concerns as follows-

1. Authors does not mentioned the number of sample replicates per sampling they have done. Generally three replicates are required for any such study.

2. From their methodology it is not clear that they have calculated the relative abundance of the microbial texa from the raw reads or the assembled reads. This needs to be clarified.

3. The authors must appreciate the requirements of the profile of some important chemicals like nitrate, nitrite, sulfate etc. in the waters for corelating microbial texa of definite function within the biome

4. Some extended study to validate the alive and active microbial presence in the metagenome is required like metatranscriptomics or isolation and characterization of some abundant microbes as per metagenomics.

5. In the discussion the authors should add a section for the possible functionalities and possibility of any change of that through out the study period of the microbiomes as appeared from this study.

6. PLOS authors have the option to publish the peer review history of their article (what does this mean?). If published, this will include your full peer review and any attached files.

Reviewer #1: No

Reviewer #2: **Yes: **Sabyasachi Bhattacharya

---

## [Author Response · Author response to Decision Letter 0]

11 Aug 2023

Dear Dr. Mapder, 

Thank you kindly for your message and many thanks to the reviewers as well for taking the time to go through our submission and provide valuable feedback that is sure to improve the quality of the manuscript. A point-by-point response to the additional requirements that were requested, as well the reviewers’ comments is provided below.

The formatting of the submitted material has been checked to ensure that it complies with PLOS ONE’s style requirements.

Given that the Kuwait Institute for Scientific Research (KISR) is a government research institution, no permits are required for acquisition of the seawater samples collected for the described work. All field work and sampling activities are approved once the proposal is approved and processed within the KISR system. However, our sampling team was required to contact the Kuwaiti Coast Guard (+965 1880888) prior to each sampling to inform them of the impending expedition. The Kuwaiti Coast Guard require the following information: the vessel name, company/institution name, names of the crew members, and the desired destination (in the case of this work, Stations A, K6, and 18; GPS coordinates are provided in Tables 1 and S1).

All repository information pertaining to the data relevant to this work has already been disclosed. The data has all been deposited on the NCBI database and the Data Reporting section includes the Bioproject number, the accession numbers, and a URL to all the relevant data. The Data Reporting section can be found on lines 702-705 in the updated manuscript. Additionally, the Data Availability statement included in our initial submission reads: “Yes - all data are fully available without restriction”.

We require you to either (1) present written permission from the copyright holder to publish these figures specifically under the CC BY 4.0 license, or (2) remove the figures from your submission.

The figure in question has been removed from the manuscript and replaced with a map of the sampling locations that was generated by Dr. Takahiro Yamamoto, our colleague at the Kuwait Institute for Scientific Research. The figure legend has been updated accordingly (line 129-133 in the updated manuscript).

Reviewers' comments:

Reviewer #1: This study analyzes the bacterial and archaeal community structures present in the unique waters of Kuwait, located in the northwest Arabian Gulf. The study aims to identify resident organisms and understand the dynamic nature of their existence in an environment prone to various environmental and anthropogenic stressors. The study utilizes a shotgun metagenomics approach and analyzes variations in community structures with respect to depth, season, and location, as well as their susceptibility to changes in abundance with respect to various physicochemical parameters. The results indicate an approximately even abundance of archaeal and bacterial communities but significantly greater diversity among the bacterial population. The study provides a framework for future investigations of functional genetic adaptations developed by resident biota attempting to survive in the extreme conditions of the unique aquatic ecosystem.

Line 271: Since Rhodobacteraceae_unclassified (14.41%) does not indicate genus name, it could be avoided from the genera level result descriptions.

Line 272: Similarly Candidatus_Heimdallarchaeota unclassified (13.10%),

Pelagibacteraceae unclassified, Euryarchaeota unclassified (4.37%) could be avoided.

Line 275-281: Since the usage of short gun sequencing is to highlight the species richness, the results of species level resolution could be highlighted better: example; Candidatus Heimdallarchaeota archaeon, Alteromonas macleodii, Marinobacter hydrocarbonoclasticus,

We are very thankful to the reviewer for the positive comments. In response to the above three points from the reviewer, we prefer to keep the unclassified genera and species as part of the results description for a couple of reasons. Firstly, they exhibit a greater relative abundance compared to the annotated taxons, and secondly, removing these taxons from the results will nullify one of the major advantages of employing shot-gun sequencing, which is to capture uncultivable and unknown genera within an environment.

Line 665 – 673: Conclusion should be re-written.

Example:

In summary, the current study is groundbreaking as it is the first to use shotgun metagenomics to explore bacterial community structure and to investigate archaeal community structure in Kuwaiti territorial waters of the northwest Arabian Gulf. The findings show compelling patterns in spatiotemporal variability concerning factors such as depth, season, and location. These findings are critical to laying the foundation for future research to delve deeper into the microbial community structures within this distinct marine environment. Moreover, the results will be useful for examining the functional activities these microbial populations undertake to survive and thrive in the region's harsh conditions.

We thank the reviewer for this suggestion and have incorporated the suggested text into the Conclusion section in the updated manuscript on lines 714-722. 

Reviewer #2: The authors studied the microbial diversity of Kuwaiti territorial waters of Arabian sea by shot gun metagenomics approach at three sampling points for six months. This study is certainly appreciable for its unique sampling site. There are some concerns as follows-

We appreciate the reviewer’s positive feedback.

1. Authors does not mentioned the number of sample replicates per sampling they have done. Generally three replicates are required for any such study.

> 3 L of seawater was collected from each sampling site each month. 2x 1 L replicates were aliquoted and filtered in our laboratories. DNA was isolated from each filter paper separately, and the isolated nucleic acid from each replicate was individually stored at -20oC. One replicate was sent for sequencing while the other replicate remains frozen in our possession for use in future alternative investigations (e.g., 16S rRNA PCR, etc.) or for re-sequencing if necessary. The team strongly believes that this approach is more than sufficient to provide the data required to draw the conclusions described. An important source of the confidence in this approach is the depth of sequencing and the quality of the sequencing data generated as evident from the FASTQC files provided by the sequencing provider, which is a biotech company based in the U.S.

2. From their methodology it is not clear that they have calculated the relative abundance of the microbial texa from the raw reads or the assembled reads. This needs to be clarified.

We thank the reviewer for bringing up this important point. The relative abundance of the microbial taxa was calculated from the assembled reads. This information has been added to the manuscript on line 195 to improve clarity.

3. The authors must appreciate the requirements of the profile of some important chemicals like nitrate, nitrite, sulfate etc. in the waters for corelating microbial texa of definite function within the biome

The relevance of the reviewer’s suggestion is greatly appreciated. In the present work, data pertaining to the levels of these parameters, including levels of nutrients such as nitrate, nitrite, and sulfate, are not included. However, other ongoing projects in our department are routinely measuring the levels of these nutrients and curating the data. Thus, while such data might be beyond the scope of the present work, we will aim to incorporate such information in future analyses. That being said, we have referred to the published literature on this aspect and added citations with relevant information to the text (lines 611-615 in the updated manuscript). The information has also been added as a note under Table 1 (lines 142-143 in the updated manuscript).

4. Some extended study to validate the alive and active microbial presence in the metagenome is required like metatranscriptomics or isolation and characterization of some abundant microbes as per metagenomics.

We thank the reviewer for this valuable suggestion that would surely greatly enhance the scope of the present work. Firstly, it would be preferable to perform metatranscriptomic analysis rather than microbiological isolation given that one of the main advantages of genome-based methods in this context is to enable the capture both cultivable and non-cultivable microbes. However, given that such analyses were not included in the original scope of the present work, it will not be possible to present the results in the current manuscript. However, future efforts will aim to identify the metabolically active microbial forms using such approaches. A brief note on this topic has been added to the discussion section (lines 603-606 in the updated manuscript).

5. In the discussion the authors should add a section for the possible functionalities and possibility of any change of that through out the study period of the microbiomes as appeared from this study.

We would like to thank the reviewer for this suggestion, and we have incorporated a section to that effect (lines 748-772 in the updated manuscript).

---

## [Editor Report · Decision Letter 1]

24 Aug 2023

Diversity and spatiotemporal variations in bacterial and archaeal communities within Kuwaiti territorial waters of the Northwest Arabian Gulf

PONE-D-23-05247R1

Dear Dr. Fakhraldeen,

We’re pleased to inform you that your manuscript has been judged scientifically suitable for publication and will be formally accepted for publication once it meets all outstanding technical requirements.

Kind regards,

Tarunendu Mapder, Ph.D.

Academic Editor

PLOS ONE
---

## [Editor Report · Acceptance letter]

30 Aug 2023

PONE-D-23-05247R1 

Diversity and spatiotemporal variations in bacterial and archaeal communities within Kuwaiti territorial waters of the Northwest Arabian Gulf 

Dear Dr. Fakhraldeen:

I'm pleased to inform you that your manuscript has been deemed suitable for publication in PLOS ONE. Congratulations! Your manuscript is now with our production department. 

Kind regards, 

on behalf of

Dr. Tarunendu Mapder 

Academic Editor

PLOS ONE